# Impacts of Antiretroviral Therapy on the Oral Microbiome and Periodontal Health of Feline Immunodeficiency Virus-Positive Cats

**DOI:** 10.3390/v17020257

**Published:** 2025-02-13

**Authors:** Laura Bashor, Jennifer E. Rawlinson, Christopher P. Kozakiewicz, Elisa Behzadi, Craig Miller, Jeffrey Kim, Megan Cierzan, Mary Nehring, Scott Carver, Zaid Abdo, Sue VandeWoude

**Affiliations:** 1Department of Microbiology, Immunology, and Pathology, Colorado State University, Fort Collins, CO 80523, USA; 2Department of Clinical Sciences, Colorado State University, Fort Collins, CO 80523, USA; jennifer.rawlinson@colostate.edu (J.E.R.);; 3W.K. Kellogg Biological Station, Michigan State University, Hickory Corners, MI 49060, USA; 4Department of Integrative Biology, Michigan State University, East Lansing, MI 48824, USA; 5Ecology, Evolution, and Behavior Program, Michigan State University, East Lansing, MI 48824, USA; 6Department of Veterinary Pathobiology, College of Veterinary Medicine, Oklahoma State University, Stillwater, OK 74078, USA; 7Comparative Medicine Research Unit, School of Medicine, University of Louisville, Louisville, KY 40208, USA; 8Odum School of Ecology, University of Georgia, Athens, GA 30602, USA; 9Center for the Ecology of Infectious Diseases, University of Georgia, Athens, GA 30602, USA; 10Department of Biological Sciences, University of Tasmania, Hobart, TAS 7001, Australia

**Keywords:** feline immunodeficiency virus (FIV), periodontitis, gingivitis, oral microbiome, antiretroviral therapy

## Abstract

Feline immunodeficiency virus (FIV) is the domestic cat analogue of HIV infection in humans. Both viruses induce oral disease in untreated individuals, with clinical signs that include gingivitis and periodontal lesions. Oral disease manifestations in HIV patients are abated by highly effective combination antiretroviral therapy (cART), though certain oral manifestations persist despite therapy. Microorganisms associated with oral cavity opportunistic infections in patients with HIV cause similar pathologies in cats. To further develop this model, we evaluated characteristics of feline oral health and the oral microbiome during experimental FIV infection over an 8-month period following cART. Using 16S rRNA sequencing, we evaluated gingival bacterial communities at four timepoints in uninfected and FIV-infected cats treated with either cART or placebo. Comprehensive oral examinations were also conducted by a veterinary dental specialist over the experimental period. Gingival inflammation was higher in FIV-infected cats treated with placebo compared to cART-treated cats and the controls at the study endpoint. Oral microbiome alpha diversity increased in all groups, while beta diversity differed among treatment groups, documenting a significant effect of cART therapy on microbiome community composition. This finding has not previously been reported, and indicates cART ameliorates immunodeficiency virus-associated oral disease via the preservation of oral mucosal microbiota. Further, this study illustrates the value of the FIV animal model for investigations of mechanistic associations and therapeutic interventions for HIV’s oral manifestations.

## 1. Introduction

Feline immunodeficiency virus (FIV) is a common and incurable infection of domestic cats and a disease analogue to HIV in humans [1,2]. Untreated FIV and HIV infections induce oral dysbiosis in the host and commonly result in oral manifestations [3,4,5]. Oral disease can manifest as oral candidiasis (“thrush”), linear gingival erythema, necrotizing ulcerative gingivitis, and necrotizing ulcerative periodontitis in humans, and as the analogous feline chronic gingivostomatitis (FCGS) in cats. Each of these disease manifestations are characterized by a shift in the community composition of the oral microbiome. Changes in the feline oral microbiome associated with FCGS have been documented [6,7], but studies elucidating the specific effects of FIV are contradictory and require further study [8,9].

Highly effective combination antiretroviral therapy (cART) for humans living with HIV has significantly decreased the incidence of oral manifestations; however, oral disease manifestations continue to be documented, particularly in children [10,11,12,13,14,15]. It is unknown whether the ameliorating impact of cART on oral disease is related to a direct antiviral effect, is a byproduct of stabilized oral immune function, or results from a combination of these or other factors. Gingivitis and other oral diseases are the most common symptom of chronic FIV-infection in naturally infected domestic cats [1,16,17,18]. While FIV-infected cats have been treated less commonly with antiviral drugs, several potentially effective regimens have been documented [19,20,21,22]. Therefore, we investigated the impacts of cART on the oral microbiome and oral health to aid hypothesis generation about potential mechanisms, and as a prelude to potential therapeutic interventions.

We performed detailed examinations of the oral cavities of cats following experimental infection with FIV in conjunction with 16S rRNA sequencing of gingival biopsies to investigate oral microbial community structure. Cats were divided into three experimental groups (healthy uninfected (Control), FIV+ placebo-treated (Placebo), and FIV+ cART-treated (cART)). We hypothesized that this study design would enable longitudinal comparisons among the three treatment groups that would disentangle the independent effects of FIV and antiretroviral therapy on feline oral health and the oral microbiome.

## 2. Materials and Methods

### 2.1. Study Population

Study animals were maintained within a research cat colony at Colorado State University, an Association for Assessment and Accreditation of Laboratory Animal Care International (AAALAC)-accredited institution. The study design was reviewed and approved by the Colorado State University Clinical Research Review Board and Institutional Animal Care and Use Committee (IACUC; protocol #1142, approved 22 July 2020). Animals used for the study were part of a larger research study aimed at determining if cART could be a viable treatment for cats infected with FIV [22].

A total of 18 domestic shorthair cats were enrolled in the study and divided into three groups. Groups were matched to distribute age, gender, sex, and littermates randomly. The sex distribution included 9 intact females and 9 neutered male cats. Study cats ranged in weight from 2.8 to 4.4 kgs and 26 to 39 weeks (6.5 to 10.5 months) old at the start of the study and 60 to 73 weeks (15 to 18.25 months) old at the end of the study. Cats included in the study were clinically healthy with complete adult dentition. Exclusion criteria included incomplete dentition, history of oral trauma, dental malformations that would inhibit periodontal scoring, and oral mucosa inflammation. Cats were also excluded from the study if antibiotics or anti-inflammatory medications had been administered within 30 days of the study start date.

All cats were housed in a communal living situation of 3–4 cats per housing unit and received the same commercially available diet (IAMS Proactive Health). Various commercially available wet foods were used as positive reinforcement during cART and placebo administration (Nestlé PURINA Fancy Feast and Friskies, INABA Foods Churu). Once infected, FIV+ cats were housed separately from control cats. All cats were specific pathogen free (SPF) and raised in the same SPF colony setting for the duration of the study. All cats were monitored regularly during the day, with attention paid to individual eating, energy levels, grooming behavior, oral comfort, and body condition. Physical examinations including weight, temperature, heart rate, respiratory rate, and mentation were performed weekly during weeks 0–6, then every other week for the remainder of the study period. At the end of the study, animals were returned to the research colony or were adopted out. No animals were euthanized for this study.

### 2.2. FIV Inoculation and Treatment Groups

Three groups of six cats each were devised for the study: healthy uninfected (Control), FIV+ placebo-treated (Placebo), and FIV+ cART-treated (cART). At week 0, 12 of the 18 cats selected for the study were inoculated intravenously (IV) and orally with FIV_C36_ viral stock diluted 1:80 with 0.9% sterile saline, a molecular clone of FIV that is immunopathogenic and induces reproducible oral lesions [23,24,25,26]. The six cats in the Control group were sham inoculated with 1 mL of 0.9% saline. Beginning at week 5 post-inoculation through week 24, six FIV+ cART cats were treated with daily cART administered subcutaneously consisting of two reverse transcription inhibitors, tenofovir disoproxil fumarate (PMPA, 20 mg/kg/day, donated by ViiV Healthcare) and emtricitabine (FTC, 40 mg/kg/day, donated by Gilead), and an integrase inhibitor dolutegravir (DTG, 2.5 mg/kg/day, donated by Gilead). The remaining six FIV+ Placebo cats received 1 mL daily subcutaneous injections of 15% kleptose, the cART vehicle. Infection was confirmed in all twelve FIV-infected cats via digital droplet PCR, and the viral infection dynamics, drug absorption, and hematologic parameters observed during the study have been described elsewhere [22].

### 2.3. Oral Health Data and Sample Acquisition

All cats were sedated with a similar anesthetic protocol that included Ketamine 2 mg/kg (100 mg/mL), Butorphanol 0.2 mg/kg (0.3 mg/mL), and Dexdomitor 0.01 mg/kg (0.5 mg/mL). A blinded veterinary dental specialist performed all oral examinations and sample collections. Once sedated, full mouth intraoral radiographs were acquired using a size 2 digital radiography sensor connected to a computer with digital imaging software. All images were stored in Digital Imaging and Communications in Medicine (DICOM) format on the acquisition computer. Intraoral and dental photographs were acquired of all regions of the mouth. Radiographs and regions photographed were evaluated at the time of collection for inflammation and signs of attachment loss. Radiographs were used to supplement and confirm findings on probing.

A complete extraoral-intraoral examination was performed. Bilateral mandibular lymph nodes were palpated, and the level of reaction was recorded as normal (no enlargement, 0), mild (slight increase in size and firmness of one lymph node, 1), moderate (marked increase in size of two lymph nodes, 2), or severe (2 or more lymph nodes palpating larger than mandibular salivary gland, 3). Intraoral examination was performed with a UNC-15 periodontal probe and #23 explorer. Overall oral inflammation was scored with the Stomatitis Disease Activity Index (SDAI) by the veterinary dental specialist, and technical staff responsible for daily care of the animals reported historical “owner” information for the SDAI [27]. The Total Mouth Periodontal Scoring system was used to assess gingivitis (measured as TMPS-G) and attachment loss (measured as TMPS-AT) resulting from periodontitis [28,29,30,31]. Gingivitis was measured on the Silness–Löe index of 0–3 [32]. Briefly, a 0 gingival index was applied for no visible gingival inflammation or bleeding upon probing; a 1 gingival index was applied for gingival erythema or edema with no bleeding upon probing; a 2 gingival index was applied for gingival erythema or edema with mild gingival bleeding upon probing; a gingival index of 3 was applied for gingival erythema/edema with spontaneous bleeding without probing. As per the TMPS protocol, a subset of teeth was measured on the buccal aspect. All measurements were made on the left side of the mouth. Measurements for attachment loss were recorded in 0.5 mm increments. When the periodontal probe could not be placed into the gingival sulcus, the attachment was measured as 0 mm. The above procedure was repeated for weeks −1, 5, 11, 16, 24, and 33.

Additional samples were collected for characterization of the oral microbiome at specific time points throughout the study. At weeks −1, 5, 11, and 24, a 5 mm (length) by 1 mm (width) strip of gingiva was harvested from the palatal/lingual free gingival margin of a canine tooth with a #11 scalpel blade prior to radiographs and periodontal probing. For each collection time a different canine tooth was used. Samples were aseptically collected in sterile tubes, placed in a cooler with dry ice for transport, and frozen at −80 degrees Celsius until the microbiome analysis could be performed. After all data was collected, the cats were reversed with Antisedan 0.01 mg/kg (0.3 mg/mL). Cats recovered quickly from sedation without complications and were returned to colony housing. At week 24 a benign mass was observed on Cat 11’s tonsil and was removed.

### 2.4. Oral Health Analyses

Mixed effects Generalized Additive Models (GAM) were used to evaluate how oral health variables (gingival inflammation, attachment loss, submandibular lymph node size) differed among the treatment groups (Control, cART, Placebo) on average and over time, with cat ID as a random effect to accommodate the repeated measures. GAMs were specifically selected in this study to account for the frequent non-linear temporal dynamics in response variables and to provide superior capacity to evaluate treatment differences over time compared to common linear methods (e.g., repeated measures ANOVA). Within each GAM, treatment was included as a factorial fixed effect to examine average differences. An interaction term between treatment and time (week) as a spline factor was also included to evaluate whether each treatment group changed over time. A post hoc test was used to determine whether the temporal splines differed among the treatment groups. Prior to all analyses, the distributions of the response variables were assessed and log10 transformed to normalize the data. All analyses were undertaken in Rv4.0.3 using the packages ‘mgcv’ and ‘car’. Results of statistical analyses are presented in Table 1.

### 2.5. DNA Isolation and 16S Amplicon Sequencing

DNA was isolated from gingival biopsy samples using a modified protocol for the DNeasy Blood & Tissue Kit (Qiagen, Germantown, MD, USA) with added homogenization steps. Biopsies were incubated in 270 µL Buffer ATL (Qiagen) in Lysing Matrix C tubes (MP Biomedicals, Irvine, CA, USA) with 30 µL Proteinase K (Qiagen) at 56 °C for 2 h. Following incubation, samples were homogenized for 30 s at 6.5 m/s in a FastPrep instrument (MP Biomedicals). Following homogenization, another 30 µL Proteinase K was added and samples were incubated at 56 °C overnight. The remaining steps were carried out according to the Qiagen DNeasy Blood & Tissue Kit protocol using 200 µL of lysate. DNA was eluted twice in 50 µL of Buffer EB (Qiagen), preheated to 70 °C with a 3-min incubation each time, for a final volume of 100 µL. Three blank extractions were carried out for negative controls. Gingival biopsy DNA samples (*n* = 72 samples, 12 ng/µL aliquots), three negative control samples, and three Zymo microbial community positive controls were prepared for 16S rRNA amplicon sequencing at Novogene (Davis, CA, USA; *n* = 78 samples total).

### 2.6. Bioinformatics Analyses

Next-generation sequencing datasets were subjected to quality control and trimming (Trimmomatic) prior to analysis with mothur software version 1.48.0 following a modified MiSeq SOP: https://mothur.org/wiki/miseq_sop/ (accessed on 18 September 2022) [33]. For the mothur analysis, data were reduced to unique sequences and their associated frequencies. The unique sequences were then classified using the SILVA taxonomic database version 138.1 [34], obtained from https://mothur.org/wiki/silva_reference_files/ (accessed on 30 September 2022). Chimeric sequences and other potential contaminants (such as mitochondria) were removed. Data were then clustered into operational taxonomic units (OTUs) using distance-based greedy clustering at 97% identity and reclassified to identify the prevalent microbial taxa and the associated abundances. Positive controls were used to apply a cutoff to the data based on the known composition of the positive control to prevent overestimation of diversity and to calculate an overall sequencing error rate. After this processing, 13, 29, and 8 OTUs remained in the three negative controls (see Appendix A).

Alpha and beta diversity calculations and data visualization were carried out in Rv4.1.2 using the packages ‘phyloseq’, ‘vegan’, and ‘metagenomeSeq’ [35,36,37]. Alpha diversity measures calculated included the Shannon Diversity index, the Inverse Simpson Diversity index (both produced by the ‘phyloseq’ package), and the Vegan-Normalized Richness (the expected richness for each sample after normalizing for sequencing depth calculated from rarefaction curves produced by the ‘vegan’ package). These diversity measures were analyzed for differences among treatment groups and over time using the same GAM approach described above. We additionally assessed if there was any relationship between Vegan-Normalized Richness with gingival inflammation or attachment loss using a Spearman correlation analysis.

To assess the relationship of microbial community structures (beta diversity) at the OTU level across treatment groups, data were normalized using cumulative sum scaling (‘metagenomeSeq’ package), analyzed with Permutational-based Multivariate Analysis of Variance (PERMANOVA) (adonis function in vegan) with cat as a random effect, and visualized with non-metric multidimensional scaling (NMDS) of the Bray-Curtis dissimilarity matrix (metaMDS function in ‘vegan’). To visualize the relative abundances of taxa at the phylum level, OTU data were filtered for OTUs with at least 10 reads detected in at least 5 samples, and relative abundance was calculated by dividing the number of reads for each OTU in a sample by the total number of reads detected in that sample.

## 3. Results

### 3.1. Assessment of Feline Oral Health

Eighteen cats were divided into three treatment groups (Figure 1): age and sex-matched healthy uninfected (Control), FIV+ placebo-treated (Placebo), and FIV+ cART-treated (cART). Viral load, hematologic parameters, and cART pharmacokinetics have been previously reported [22].

Gingival inflammation varied among treatment groups over time (Figure 2A,B, Table 1). Inflammation decreased slightly over time in the cART group, exhibited a slight increase in the Control group, and increased markedly in the Placebo group (Figure 2A,B). The temporal pattern of gingival inflammation in cART-treated cats differed significantly from the Control and Placebo groups (Table 1). At week 33, we observed lower inflammation in cART cats compared to Placebo cats (Figure 2A,B). Attachment loss also varied among treatment groups, increasing in all groups over time, but with the most marked increase occurring in the Placebo group (Figure 2C,D). Attachment loss in the Placebo group increased more than in the cART group (Table 1), and at week 33 we observed significantly lower attachment loss in cART cats compared to Placebo cats (Figure 2C).

We also measured oral cavity inflammation and submandibular lymph node reactivity. Oral cavity inflammation, quantified by the Stomatitis Disease Activity Index (SDAI), remained at baseline (SDAI = 2) for all animals at all timepoints in the study. However, we noted a time by treatment relationship in submandibular lymph node size, with both left and right lymph nodes exhibiting the same relationship (Table 1, see Figure 2E,F for the left lymph node). Submandibular lymph node size increased overall in all three groups. In contrast to the Control and Placebo groups, the cART group exhibited a marked increase in lymph node size to week 11 and decreased thereafter (Table 1, Figure 2E,F).

### 3.2. 16 rRNA Sequencing and Alpha Diversity of the Oral Microbiome

To evaluate the diversity and structure of the oral microbiome, gingival biopsies were collected at weeks −1, 5, 11, and 24 post-inoculation and subjected to 16S rRNA sequencing. Next-generation amplicon sequencing datasets were recovered from 72 feline gingival biopsies, three negative controls, and three positive controls (*n* = 78 samples total) with sufficient coverage depth for OTU clustering (Appendix A). After quality control, the median sequencing depth for non-control samples was 47,769 sequences (range: 9223–85,797) with a median of 49 OTUs per sample (range: 25–105). A total of 37 unique OTUs were observed in the three negative control samples (Appendix A). The overall sequencing error rate was 0.000465%.

Three measures of alpha diversity were calculated using the OTU counts for each sample: the Shannon Diversity index, Inverse Simpson Diversity index, and Vegan-Normalized Richness (Appendix A). All measures showed a similar overall increase among the treatment groups (Table 1). Alpha diversity did not differ between the Control and cART groups, whereas the Placebo group differed over time (by most measures), exhibiting increased alpha diversity around week 11 and declining to similar levels as the Control and cART groups by week 33 (Table 1, Figure 3 and Appendix A).

We also evaluated if there was a relationship of alpha diversity with gingival inflammation and attachment loss, finding no correlation (Spearman correlations: r = 0.07, *p* = 0.556 and r = 0.07, *p* = 0.583, respectively).

### 3.3. Beta Diversity of the Oral Microbiome

We detected a diverse oral microbiome across individuals in this study, with distinct variation among individuals, across treatment groups, and over the course of the study (Figure 4). The phyla with the highest average relative abundance across all three treatment groups at all time points were *Proteobacteria*, *Bacteroidetes*, *Fusobacteria*, *Actinobacteria,* and *Firmicutes* (Figure 5). At weeks 1 and 5 post-inoculation, *Proteobacteria* were dominant, with mean relative abundances between 0.68 and 0.79 in all three treatment groups, whereas in weeks 11 and 24 of the study the predominant phyla were more evenly distributed among *Proteobacteria*, *Bacteroidetes*, and *Fusobacteria.*

Although we did not observe differences among treatment groups at the level of alpha diversity at the end of the study, we did find differences in beta diversity in samples from different treatment groups and study weeks (Figure 6). Both treatment group and the interaction between treatment group and week had a significant effect on microbial community structure at the OTU level (Table 1). Pairwise comparisons indicated that differences in microbial community structure over time existed among all treatment groups (Table 1), as illustrated by the differing trajectories of community structure (Figure 6).

## 4. Discussion

FIV infection of domestic cats has many parallels to HIV infection in humans, and the development of oral dysbiosis is a characteristic of both of these diseases. Although antiretroviral therapies are ubiquitous in human medicine and have been described in felids, few studies have considered oral health and disease in combination with these therapies. To address this gap, we evaluated the oral health and microbiomes of domestic cats experimentally infected with FIV, with and without combination antiretroviral therapy consisting of two reverse transcription inhibitors, tenofovir disoproxil fumarate and emtricitabine, and the integrase inhibitor dolutegravir.

The most notable findings of our study are specific differences in feline oral health and microbiome community structure associated with cART treatment. In particular, the differences in gingival inflammation and attachment loss among treatment groups indicate a beneficial effect of cART on the oral health of FIV+ cats. The alpha diversity of the oral microbiome in cART cats also followed a similar temporal pattern as that of uninfected control cats and was distinct from the Placebo group. Furthermore, overall microbial community structure differed significantly among all three treatment groups over the course of the study. These findings may have been strengthened by the addition of a treatment group of uninfected cats receiving cART; however, we were still able to detect a statistically significant effect of antiretroviral therapy. Future studies will benefit from directly assessing the effects of cART on the oral health of uninfected cats.

Although the majority of microbial taxa observed across all three treatment groups were consistent with previous reports of the feline microbiome [6,7,8,9], these taxa were detected in different proportions in the three treatment groups at the final time point assessed. This result indicates that distinct shifts in gingival microbial community structure occurred over the course of the study associated with FIV infection and treatment. Interestingly, we found no discernible effect of FIV infection alone (Placebo vs. Control) on oral microbiome diversity or composition. It is possible that alterations in the oral microbiome that have been associated with untreated FIV infection in the past [7,9] do not develop prior to the establishment of the chronic stage of disease. During this stage, immunosuppression and ongoing periodontal disease can cause significant alterations in oral microbiome diversity and composition due to the overgrowth of opportunistic microorganisms. Thus, a logical future direction is to extend this research to a more advanced stage of periodontal disease associated with the chronic phase of infection.

The changes in the structure of the feline oral microbiome in the presence of cART therapy observed in our study have not previously been reported. However, our findings are consistent with previous investigations of the oral and intestinal microbiomes of HIV+ individuals on antiretroviral therapy compared to HIV- individuals. One study described an antibacterial effect of ART, associated with drug category, that could lead to exacerbated dysbiosis in the human gut microbiome [38]. In another study, HIV+ subjects exhibited detectable differences in intestinal but not salivary microbiota compared to HIV- control subjects [39]. However, throat swabs from HIV+ individuals exhibited decreased microbial diversity compared with the HIV- control subjects before and after beginning retroviral therapy [40].

Two recent studies used multivariate analyses to assess the relationships between HIV infection, antiretroviral therapy and the oral microbiome, together demonstrating significant, independent effects of HIV infection and ART [41,42]. In addition to cross-sectional studies demonstrating antimicrobial effects of ART in patients, the direct testing of drugs in vitro has also been conducted. A recent study assessed the effects of twenty-five antiretroviral drugs on *Bacillus subtilis* and *Escherichia coli* growth [43]. The strongest effects were observed in the presence of Efavirenz, 2′,3′-dideoxyinosine, and zidovudine, but tenofovir disoproxil fumarate, emtricitabine, and Raltegravir (another integrase inhibitor) also exhibited antibacterial activity.

We observed significant differences in gingival inflammation, attachment loss, and submandibular lymph node size among treatment groups over the 33-week study period. Signs of gingivitis were more prominent in the Placebo group of cats, with significantly higher gingival inflammation and attachment loss compared to the cART group by the end of the study. This finding, in combination with the differences we observed in the structure of the oral microbiome, indicates that antiretroviral treatment can have a positive impact on feline oral health. We also noted increased submandibular lymph node reactivity in the cART group at week 11. Regional lymphadenopathy occurring after two months of FIV infection may be a response to active viral replication during the acute phase of FIV. Oral lymphoid tissues in particular experience increased FIV viral loads and have been proposed as a key site for FIV replication [44]. As previously reported, circulating FIV viral RNA peaked in week 2 for the Placebo group, but not until week 6 for the cART group [22]. Furthermore, viral load was reduced in the saliva but not the plasma of the cART group as compared to the Placebo group.

Interestingly, week 11 was also the timepoint at which the alpha diversity of oral microbiome in the Placebo group peaked, being significantly higher than in the cART and Control groups. This finding supports our hypothesis of a possible antibacterial effect of cART treatment. It is possible that other study variables prevented us from detecting this effect at other timepoints. Alpha diversity increased across all three treatment groups over the course of the study, most likely due to changes that occur during the first year of development as juvenile cats become more independent and permanent dentition is established [45].

The experimental design of our study controlled for environment, diet, breed, and sampling method, which, among other factors, may impact feline oral microbiome structure [46,47,48]. Nonetheless, our characterization of the bacterial microbiome is likely incomplete due to inherent limitations in 16S rRNA sequencing. Future studies could use complementary metagenomic approaches to increase taxonomic resolution, gain functional information, and identify microorganisms not captured with 16S. Evaluating oral fungal communities may be particularly valuable, as certain fungal species have previously been associated with the pathogenesis of FCGS [49].

## 5. Conclusions

Our study illustrates the changes in oral health and microbiota associated with treated and untreated multi-month FIV infection in domestic cats. Specifically, our findings demonstrate the significant influence of combination antiretroviral therapy on the gingival tissues of FIV+ cats. Antiretroviral therapy initiated early in FIV infection led to shifts in the community structure of the oral microbiome. Furthermore, the development of FIV-associated oral manifestations was only observed after 35 weeks of infection, and at this point we observed decreased gingival inflammation in the cART treatment group. This result highlights the importance of extended longitudinal studies that capture the establishment of the chronic stage of FIV infection. Furthermore, this study demonstrates the relevance of FIV as a model for HIV research.

## Figures and Tables

**Figure 1 viruses-17-00257-f001:**
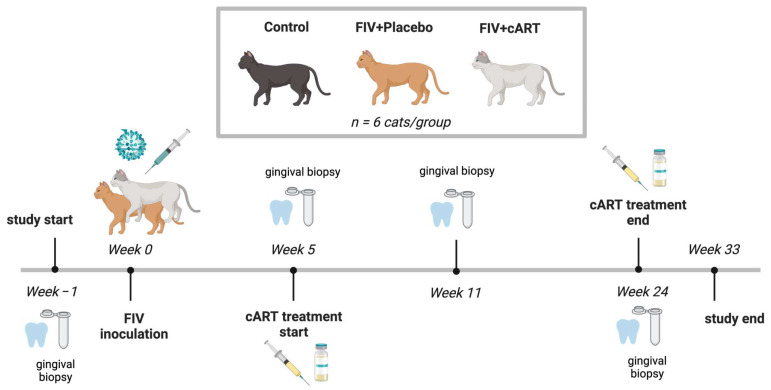
Three experimental groups of six cats each were included in the 33-week study (*n* = 18 cats total): control, FIV+ Placebo, and FIV+ cART. Created in BioRender.

**Figure 2 viruses-17-00257-f002:**
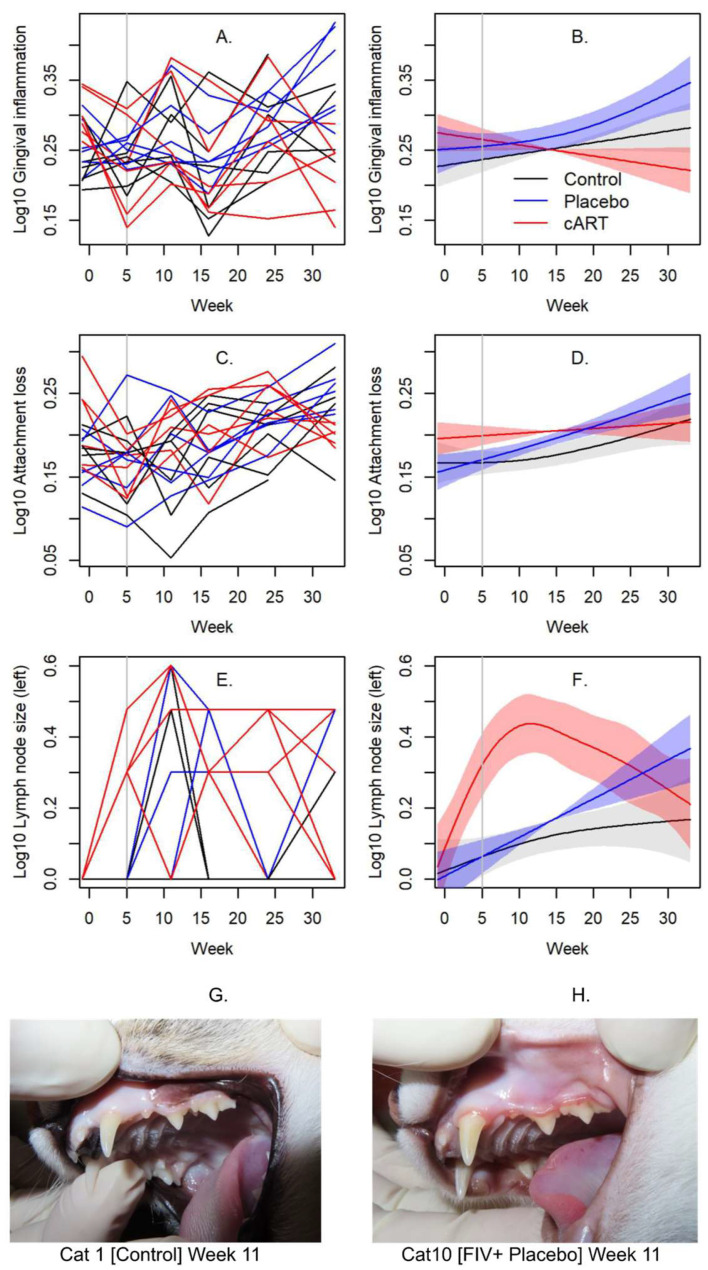
Oral health indices disrupted by FIV infection are ameliorated in the cART treatment group after 33 weeks. Oral health was assessed in three treatment groups (Control, FIV+ placebo-treated [Placebo], and FIV+ cART-treated [cART]) over the course of 33 weeks post-FIV inoculation. Gingival inflammation (**A**,**B**), attachment loss (**C**,**D**), and lymph node size (**E**,**F**) varied significantly among treatment groups over time (see Table 1 for statistics and post hoc comparisons). Oral health measures are presented as line plots (**A**,**C**,**E**) next to their respective spline fits from GAM analyses (**B**,**D**,**F**). Lines are colored by treatment (black = Control, blue = Placebo, red = cART). Shading represents standard deviations of the spline fits. The vertical line indicates when cART treatment commenced. Images contrast oral health in (**G**) sham-inoculated control (normal gingival margin), and (**H**) FIV+ placebo-treated individual (gingival inflammation) at 11 weeks post-inoculation.

**Figure 3 viruses-17-00257-f003:**
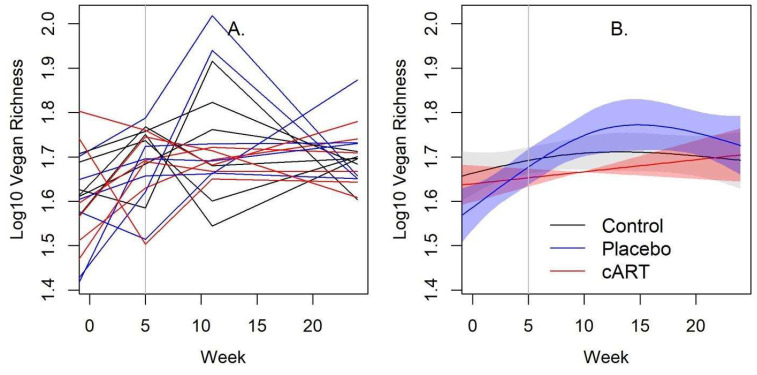
The temporal pattern of alpha diversity of the gingival microbiome in cART-treated cats more closely resembled uninfected control cats, and both differed from the FIV+ placebo-treated group. The normalized richness of the gingival microbiome at study weeks −1, 5, 11, and 24 post-inoculation is shown for three treatment groups of domestic cats: uninfected control (Control), FIV+ placebo-treated (Placebo), and FIV+ cART-treated (cART) (*n* = 72 samples, six per treatment group). Expected richness for each sample normalized by the minimum sequencing depth was calculated on the OTU level from 16S rRNA sequencing data using the rarefy() function in the R package ‘vegan’. In plot (**A**), each line is an individual cat, colored by treatment (black = Control, blue = Placebo, red = cART). In plot (**B**), the associated spline fit from GAM analysis is presented, with shading representing standard deviations. The vertical line indicates when cART treatment commenced.

**Figure 4 viruses-17-00257-f004:**
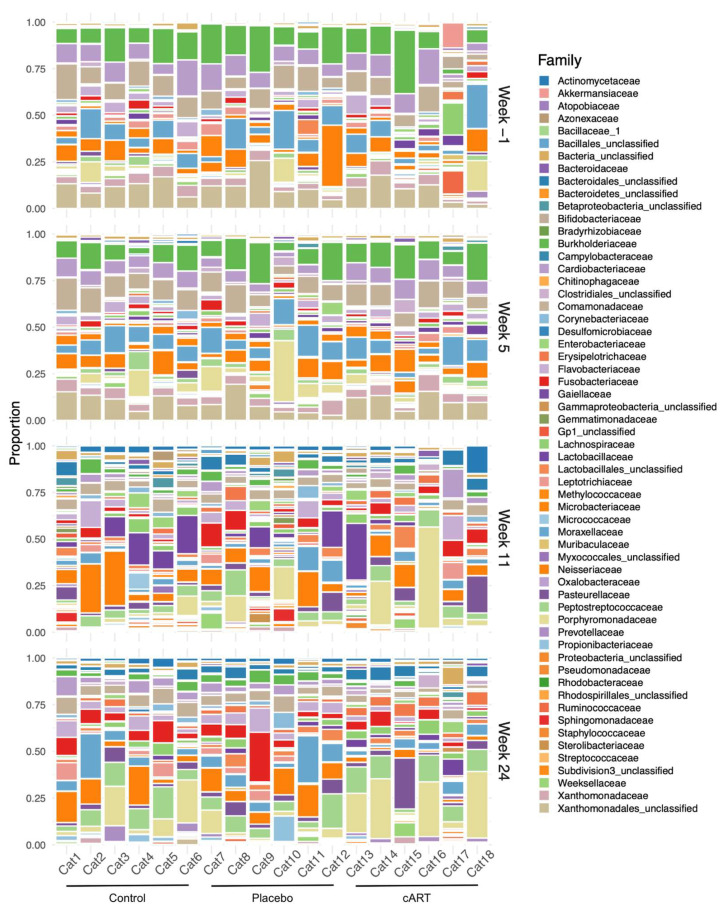
The composition of the gingival microbiome varied among individual cats, across treatment groups, and over time. Bacterial families observed in three treatment groups of domestic cats (uninfected control (Control), FIV+ placebo-treated (Placebo), and FIV+ cART-treated (cART); 6 cats/group), over the course of a six-month study are shown. Samples were collected from all individuals at weeks −1, 5, 11, and 24 for a total of *n* = 72 samples. Each bar represents the non-normalized proportions of bacterial families detected in greater than 1% of a sample. Each sample was generated from a single gingival biopsy collected from one cat at one time point and then subjected to 16S rRNA sequencing.

**Figure 5 viruses-17-00257-f005:**
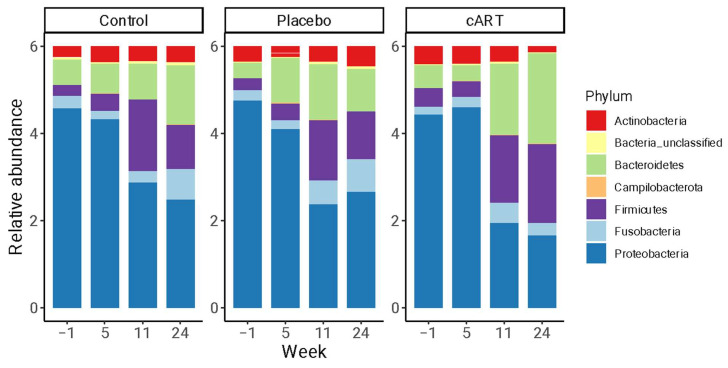
Oral microbiomes were dominated by *Proteobacteria*, *Bacteroidetes*, *Fusobacteria*, *Actinobacteria,* and *Firmicutes* across all three treatment groups at all time points. Relative abundances are shown for bacterial phyla identified by 16S rRNA sequencing of gingival biopsies collected from three treatment groups of domestic cats (uninfected control (Control), FIV+ placebo-treated (Placebo), and FIV+ cART-treated (cART); 6 cats/group), over four timepoints (study weeks −1, 5, 11, and 24) (*n* = 72 samples). Each bar presents all bacterial phyla detected in greater than 1% of a sample for one treatment group of cats at one time point.

**Figure 6 viruses-17-00257-f006:**
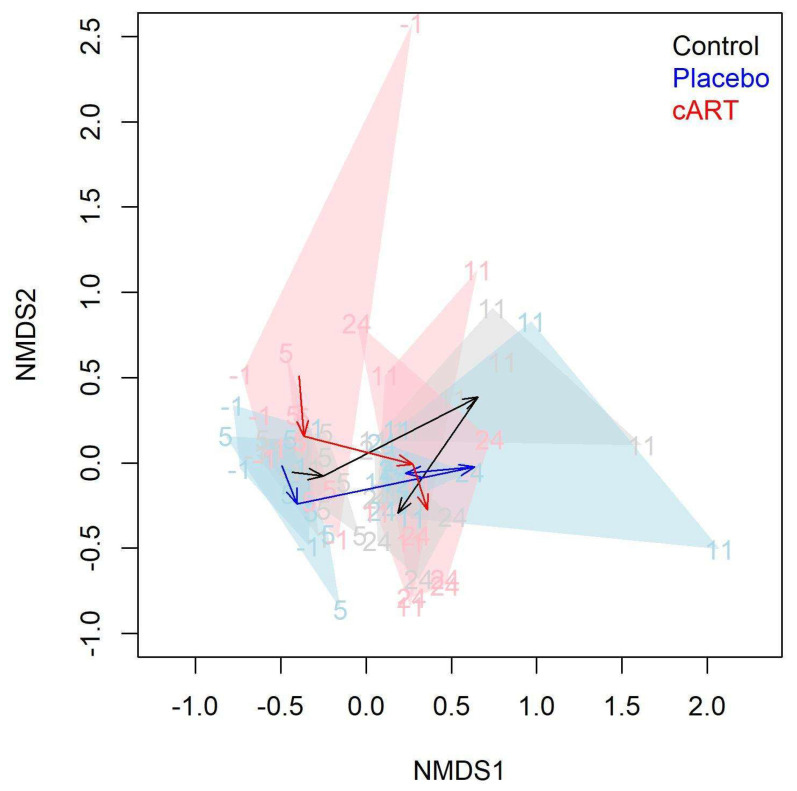
The beta diversity of the oral microbiome differed significantly among treatment groups over time. Variation in gingival 16S microbial community structure at the OTU level is shown for three treatment groups of domestic cats (Control, FIV+ placebo-treated (Placebo), and FIV+ cART-treated (cART)) over four timepoints (study weeks −1, 5, 11, and 24) (*n* = 72 samples). Data are visualized in two dimensions with non-metric dimensional scaling (NMDS) of the Bray–Curtis dissimilarity matrix. Numbers and polygons show groupings of treatment and time groups. Arrows represent the centroid of each treatment group over time and can be interpreted as average trajectories of community change.

**Table 1 viruses-17-00257-t001:** Measures of oral health and gingival microbiome alpha and beta diversity differed significantly among treatment groups over the course of the study. Mixed effects Generalized Additive Models testing if (1) variables differ among Treatment groups (Control, Placebo, cART) on average; (2) variables change over time for each Treatment group (spline-fit of interaction term between Treatment and Week); and (3) post hoc comparisons testing if the temporal dynamics of variables differ among Treatment groups, represented by letters (a, b, c) adjacent to *p*-values. Bolding denotes cases where post hoc comparisons identify Treatments to differ over time. Model deviance is also given. PERMANOVA is also presented, performing a comparable analysis for beta diversity. Note that PERMANOVA considers linear effects of time on microbial community structure only, so an overall fixed Week * Treatment interaction was included, followed by post hoc testing of the interaction among Treatment pairs.

		Treatment	sWeek * Control	sWeek * Placebo	sWeek * cART	Deviance (%)
Oral Health						
Gingival	F	0.99	3.82	7.13	4.41	47.5
Inflammation	P	0.376	**0.054 a**	**<0.001 a**	**0.039 b**	
Attachment Loss	F	0.75	3.78	19.1	3.22	64.7
	P	0.476	**0.026 ab**	**<0.001 a**	**0.015 b**	
Lymph Node Size	F	6.62	5.30	15.38	9.95	64.5
Right	P	0.002	**0.024 a**	**0.002 a**	**<0.001 b**	
Lymph Node Size	F	5.79	2.03	5.98	7.31	57.7
Left	P	0.004	**0.121 a**	**<0.001 a**	**<0.001 b**	
Microbiome						
Shannon Diversity	F	1.17	2.26	7.25	5.08	36.1
	P	0.318	**0.138 a**	**0.001 b**	**0.028 a**	
Inverse Simpson	F	1.75	0.63	3.64	0.70	17.8
Diversity	P	0.181	0.431 a	0.029 a	0.408 a	
Vegan Richness	F	0.31	0.67	6.23	1.69	28.6
	P	0.733	**0.448 a**	**0.002 b**	**0.198 a**	
PERMANOVA	F	1.08	Week * Treatment: F = 7.70, *p* < 0.001			27.7
	P	<0.001	**a**	**b**	**c**	

## Data Availability

The original contributions presented in this study are included in the article/Appendix A. Analysis scripts and data tables are available on Github at https://github.com/laurabashor/FIV_oral_16S (accessed on 21 February 2024). The 16S sequencing data presented in the study are openly available in the National Center for Biotechnology Information (NCBI) Sequence Read Archive (SRA) database under BioProject PRJNA1076586. Further inquiries can be directed to the corresponding author.

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
