# Peer review of "Impacts of Antiretroviral Therapy on the Oral Microbiome and Periodontal Health of Feline Immunodeficiency Virus-Positive Cats"

_viruses, 2025, doi:10.3390/v17020257_

Round 1

Reviewer 1 Report

Comments and Suggestions for Authors

The manuscript by Bashor and collaborator, aims to explore the potential effect of the FIV infection +/- combination antiretroviral therapy (cART), on the oral disease and oral microbiome in cats. The study is interesting and relevant, and it could highlight the importance of the FIV as a valuable model for studying HIV. In general, the manuscript is well written and the results are of relevance to the field.

In the study, the authors compared 3 experimental conditions: non-infected (control), FIV infected + cART, and FIV infected + placebo (no cART). The study is mostly well designed, however, in my opinion the authors missed one very important experimental condition in their study: non-infected + cART treatment. In the study, the results clearly described the detrimental effect of the FIV infection on the oral health. Interestingly, the cART treatment showed a decrease in oral pathology, demonstrating the potential benefit of the therapy. However, my biggest critic to the study comes in the second part, where they evaluated the effect of the experimental conditions on the oral microbiome. In this part, the effect of the cART must be evaluated with and without infection, to determine if the cART itself could alter the oral microbiome. In fact, in Figure 4 it can be observed that the three experimental groups had major differences in microbial diversity. While these differences can be observed individually in each cat, it seems to be trend in some microbial populations, especially when comparing Control vs FIV+cART. This raises questions about the potential effect of the drugs on oral microbial populations, which unfortunately, cannot be assessed in this study. I want to emphasize that the lack of this experimental group while important, does not diminish the relevance of the results, nor the importance of the study, but unfortunately it rises questions that would have been important to address, and an explanation of why this group was not considered/included in the study needs to be mentioned and discussed in the document.

My second major concern with the manuscript, is the lack of testing for FIV infection after inoculation and during the study. While the clinical manifestations are compatible with the FIV infection, there is no mention in the manuscript whether the animals were tested for infection during any point in the study. If this information is available, it must be included in the manuscript. If it is not available, I believe it is important to show that the animals were in fact infected, so every possible effort must be made to test for infection using the available samples or DNA from the sampling phase of the study.

General comments:

-              There is no mention in the manuscript about the outcome for the experimental animals after the study ended. This must be included.

-              In the abstract, it is mentioned that the “study illustrates the value of the FIV animal model for investigations of mechanistic associations and therapeutic interventions for HIV oral manifestations”. However, this is not addressed in the discussion or conclusions. I consider that one of the strengths of the study is the relevance of FIV as model for HIV research, so this should be discussed further and highlighted in the text.

-              The first paragraph of the discussion is the journal’s description of what should be included in this section. It was probably left in the text by mistake but has to be removed.

Author Response

Response to Reviewer 1 Comments

Thank you to the reviewer and editors for these helpful comments. We have responded point-by-point and refer to lines in the revised manuscript where edits have been incorporated. Changes to the original manuscript are highlighted in yellow in the revised text.

Comments 1: In the study, the authors compared 3 experimental conditions: non-infected (control), FIV infected + cART, and FIV infected + placebo (no cART). The study is mostly well designed, however, in my opinion the authors missed one very important experimental condition in their study: non-infected + cART treatment. In the study, the results clearly described the detrimental effect of the FIV infection on the oral health. Interestingly, the cART treatment showed a decrease in oral pathology, demonstrating the potential benefit of the therapy. However, my biggest critic to the study comes in the second part, where they evaluated the effect of the experimental conditions on the oral microbiome. In this part, the effect of the cART must be evaluated with and without infection, to determine if the cART itself could alter the oral microbiome. In fact, in Figure 4 it can be observed that the three experimental groups had major differences in microbial diversity. While these differences can be observed individually in each cat, it seems to be trend in some microbial populations, especially when comparing Control vs FIV+cART. This raises questions about the potential effect of the drugs on oral microbial populations, which unfortunately, cannot be assessed in this study. I want to emphasize that the lack of this experimental group while important, does not diminish the relevance of the results, nor the importance of the study, but unfortunately it rises questions that would have been important to address, and an explanation of why this group was not considered/included in the study needs to be mentioned and discussed in the document.

Response 1: Thank you for this comment and the thorough discussion of this limitation. We agree that having an additional experimental condition of uninfected cats + cART treatment would have provided more information to determine the effect of cART on the oral microbiome. We reviewed the literature and did not find an analysis of anti-retroviral therapy effects on the microbiome in the absence of retroviral infection in any species, and think this would be an interesting future study.  Due to logistical and funding constraints, we opted to use a placebo control group vs uninfected+cART group, and are unable to repeat the study at this time. We appreciate the reviewer’s statement that this limitation does not diminish the importance of the study. Accordingly, we modified the discussion to directly point out this limitation and recommend that future studies are needed, with this specific experimental design. We added the following text on lines 358-361: “These findings may have been strengthened by the addition of a treatment group of uninfected cats receiving cART; however, we were still able to detect a statistically significant effect of antiretroviral therapy. Future studies will benefit from directly assessing the effects of cART on the oral health of uninfected cats.”

Comments 2: My second major concern with the manuscript, is the lack of testing for FIV infection after inoculation and during the study. While the clinical manifestations are compatible with the FIV infection, there is no mention in the manuscript whether the animals were tested for infection during any point in the study. If this information is available, it must be included in the manuscript. If it is not available, I believe it is important to show that the animals were in fact infected, so every possible effort must be made to test for infection using the available samples or DNA from the sampling phase of the study.

Response 2: Thank you for highlighting an important aspect of most infection studies, that was not addressed in depth in this manuscript. This was because the focus of this manuscript was the feline oral microbiome rather than FIV infection dynamics. We did test blood, saliva, and lymph node aspirates for both viral and proviral genetic material using digital droplet PCR, and these results in addition to immunological findings, were previously published in Viruses (https://doi.org/10.3390/v15040822). We referenced this companion article throughout the manuscript on lines 59, 77, 115, 225, and 406. However, we agree that it is important to state explicitly that animal infections were confirmed. We have modified lines 112-115 in the Materials and Methods section to clarify this. It now states: “Infection was confirmed in all twelve FIV-infected cats via digital droplet PCR, and viral infection dynamics, drug absorption, and hematologic parameters observed during the study have been described [22].”

Comments 3: There is no mention in the manuscript about the outcome for the experimental animals after the study ended. This must be included.

Response 3: Thank you for this suggestion. We have added text to the Material and Methods section lines 98-99 stating: “At the end of the study, animals were returned to the research colony or were adopted out. No animals were euthanized for this study.”

Comments 4: In the abstract, it is mentioned that the “study illustrates the value of the FIV animal model for investigations of mechanistic associations and therapeutic interventions for HIV oral manifestations”. However, this is not addressed in the discussion or conclusions. I consider that one of the strengths of the study is the relevance of FIV as model for HIV research, so this should be discussed further and highlighted in the text.

Response 4: Thank you for this comment. Although we discussed our findings in relation to current HIV research on lines 343-347 and 375-393 of the discussion, we agree that we could emphasize the value of FIV as a model for HIV research in the conclusions.  Accordingly, we modified the conclusions on lines 433-434 to state: “Furthermore, this study demonstrates the relevance of FIV as a model for HIV research.”

Comments 5: The first paragraph of the discussion is the journal’s description of what should be included in this section. It was probably left in the text by mistake but has to be removed.

Response 5: Thank you for catching this error. The reviewer is correct, this is from the Viruses manuscript template, and we mistakenly left it in the text. We have deleted this paragraph.

Reviewer 2 Report

Comments and Suggestions for Authors

This is a very well written manuscript with a very good experimental design.  The sample size is small, but the study design, results and conclusions are all quite sound.  The data presents a useful comparative approach to addressing a problem in HIV+ patients that could not really be performed otherwise.  The feline model is an excellent way to evaluate the early gingival changes and temporal changes in the gingiva and oral microbiome.  The study presented here opens the door to larger studies with greater numbers of cats, more detailed sampling and perhaps even immune profiling / viremia / tissue virus changes over time.  Lines 338-342 seem to be instructions to the authors accidentally left in the draft.  Pictures of the gingival lesions, while not essential, would contribute greatly to the impact of the manuscript.

Author Response

Response to Reviewer 2 Comments

Thank you to the reviewer and editors for these helpful comments. We have responded point-by-point and refer to lines in the revised manuscript where edits have been incorporated. Changes to the original manuscript are highlighted in yellow in the revised text.

Comments 1: Lines 338-342 seem to be instructions to the authors accidentally left in the draft. 

Response 1: Thank you for catching this error. The reviewer is correct, this is from the Viruses manuscript template, and we accidentally left it in the draft. We have deleted these lines.

Comments 2: Pictures of the gingival lesions, while not essential, would contribute greatly to the impact of the manuscript.

Response 2: Thank you for this suggestion. We agree that images would make a nice addition to the manuscript. We modified Figure 2 (lines 238-249) to include two additional panels containing images taken during oral examinations contrasting normal and inflamed gingival margin in study animals.

Round 2

Reviewer 1 Report

Comments and Suggestions for Authors

The authors accepted the suggestions and improved the manuscript. I do not have any further comments.

Reviewer 2 Report

Comments and Suggestions for Authors

The manuscript is suitable for publication.  Thank you for making the suggested changes.